# END-TO-END INPUT SELECTION FOR DEEP NEURAL NETWORKS

## ABSTRACT

Data have often to be moved between servers and clients during the inference phase. This is the case, for instance, when large amounts of data are stored on a public storage server without the possibility for the users to directly execute code and, hence, apply machine learning models. Depending on the available bandwidth, this data transfer can become a major bottleneck. We propose a simple yet effective framework that allows to select certain parts of the input data needed for the subsequent application of a given neural network. Both the associated selection masks as well as the neural network are trained simultaneously such that a good model performance is achieved while, at the same time, only a minimal amount of data is selected. During the inference phase, only the parts selected by the masks have to be transferred between the server and the client. Our experiments indicate that it is often possible to significantly reduce the amount of data needed to be transferred without affecting the model performance much.

## 1 INTRODUCTION

Neural networks have successfully been applied to many domains (Bengio et al., 2013; LeCun et al., 2015). Two trends have sparked the use of neural networks in recent years. Firstly, the data volumes have increased dramatically in many domains yielding large amounts of training data. Secondly, the compute power of today's systems has significantly increased as well, particularly those of massively-parallel architectures based on graphics processing units. Those specialized architectures can be used to reduce the practical runtime needed for training and applying neural networks, which has led to the development of more and more complex neural network architectures (Krizhevsky et al., 2012; He et al., 2016; Huang et al., 2017).

Many machine learning applications require data to be exchanged between servers and clients during the inference phase. This is the case, for example, in remote sensing, where current projects produce petabytes of satellite data every year (Wulder et al., 2012; Li & Roy, 2017). The application of a machine learning model in this field to, e.g., monitor changes on a global scale, often requires the transfer of large amounts of data between the server and the client that executes the model, see Figure 1. Similarly, data have often to be transferred from clients to servers for further processing. For instance, data collected from mobile devices are transferred to remote servers to be analyzed by virtual assistants such as Alexa from Amazon, Siri from Apple, or the Google Assistant. A similar situation is given when energy-efficient microcontrollers (e.g., the Arduino Uno) are powered by batteries to collect sensor data from remote locations. Here, the transfer of data is often considered the most expensive operation due to the high power consumption caused by the transmission.

While the reduction of the training and inference runtimes have received considerable attention (Coates et al., 2013; Han et al., 2015; Gordon et al., 2018; Nan et al., 2016; Kumar et al., 2017; Xu et al., 2013), relatively little work has been done regarding the transfer of data induced by such server/client based scenarios. However, this data transfer between clients and servers can become a severe bottleneck that can significantly affect the way users leverage available data. In some cases, the necessary data transfer can be reduced based on prior knowledge (e.g., in case one knows that only certain input channels are relevant for the task to be conducted). Also, for some learning tasks, the data transfer can be reduced by extracting a small amount of expressive features from the raw data. In general, such feature based reductions have to be adapted to the specific tasks and might also lead to a worse performance compared to purely data-driven approaches.[1]

---

[1] Note that, in case the data resides on a public storage server, it is often *not* possible for the user to execute any code on the server side. This renders a manual feature extraction or the application of (parts of) a deep neural network impossible on the server side.

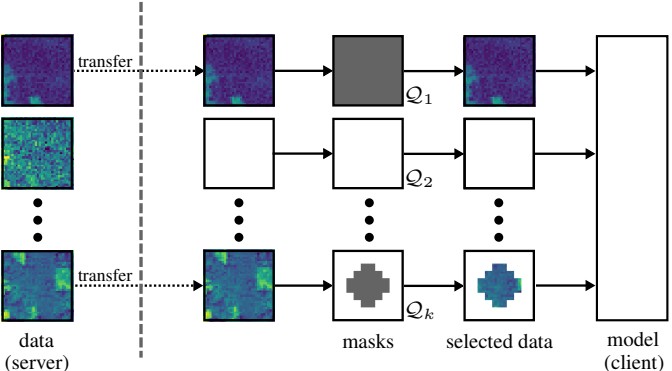

Figure 1: Application of a neural network in the context of remote sensing. Here, hundreds of input feature maps might be available (multi-spectral image data collected at different times). Transferring data from the server to the client running the model can be extremely time-consuming. Our framework uses various types of selection masks that can be adapted to the specific transfer capabilities between the server and the client (e.g., if channel- or pixel-wise data transfers are possible). Also, a different loss $\mathcal{Q}_i$ can be assigned to each individual mask to penalize selections made by it. The masks as well as the given network are optimized simultaneously in an end-to-end fashion to achieve a good model performance and to select only small amounts of the input data. During the inference phase, only the selected parts have to be transferred. Similar bandwidth-restricted scenarios can be found in other data-intensive disciplines as well such as astrophysics or in the context of sensor data analytics.

**Contribution:** We propose a framework that automatically learns to select the relevant parts of the input data for a given neural network and its task. In particular, our approach aims to select the minimal amount of data needed to achieve a model performance that is comparable with one that can be obtained on all the input data. The individual selection criteria can be adapted to the specific needs of the task at hand as well as to the transfer capabilities between the server and the client. As shown in our experiments, our framework can be used to sometimes significantly reduce the amount of data needed to be transferred during the inference phase without affecting the model performance much.

## 2 RELATED WORK

Reducing the training time has gained significant attention in recent years. This includes, for instance, the use of parallel or distributed implementations (Coates et al., 2013; Dean et al., 2012; Li et al., 2018). Approaches aiming at an efficient inference phase have been proposed as well, including schemes that aim at reducing the weights of networks or the amount of floating point operations (Han et al., 2015; Gordon et al., 2018). Similarly, methods that deploy small tree-based models have been suggested (Kumar et al., 2017; Xu et al., 2013). The transfer of data during the inference phase has been addressed as well. For instance, Nan et al. (2016) propose a method that prunes features during the construction of random forests such that only few are needed during the inference phase (thus, avoiding costs for their computation and their transfer). In some cases, data compression can be used to reduce the amount of bytes needed to be transferred (e. g., images compressed via JPEG). However, this usually requires to retrain a network to find a suitable compression level, which is not known beforehand.[2] Deep neural networks have also been used to compress image data (Jiang et al., 2018), but the resulting compressed versions are independent of the learning task.

We conduct a gradient-driven search to find suitable weight assignments for the selection masks. An alternative to our approach are greedy schemes that, e.g., incrementally select input channels or pixels. However, these schemes might yield suboptimal results since only one channel/pixel is selected in each step. Further, these approaches quickly become computationally infeasible in case many channels or input pixels are given. Naturally, an exhaustive search for finding optimal mask assignments is computationally intractable. Our approach can be seen as a trade-off between these two variants. Finally, our approach is inspired by focused and peripheral vision, where unfocused objects containing less detail still offer useful information (Strasburger et al., 2011).

---

[2]Such compressed versions might also not be available on the server/client side. Our framework can handle these scenarios as a special case with the optimal compression level automatically being selected during training.

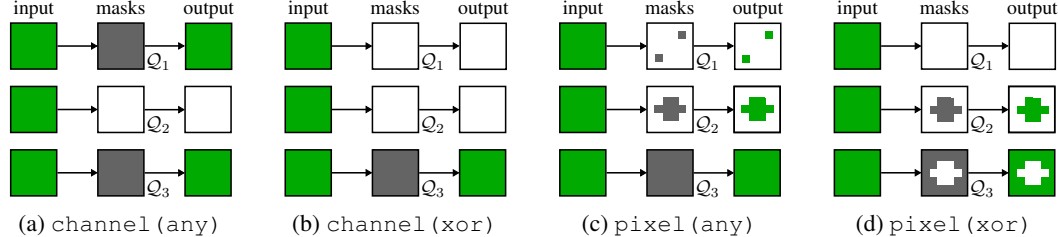

| (a) channel(any) | (b) channel(xor) | (c) pixel(any) | (d) pixel(xor) |

Figure 2: Different selection masks that can be used to select parts of the input data. For each of the masks, an individual loss $\mathcal{Q}_i$ can be defined to penalize selections made by that mask. While the final masks are discrete, differentiable surrogates are used during training.

## 3 LEARNING SELECTION MASKS

We resort to masks that can be used to select certain parts of the input data. These masks are adapted during the training process such that (a) the predictive power of the network remains satisfying and (b) only a minimal amount of the input data is selected. We will focus on image data in this work for the sake of exposition, but our approach can also be applied to other types of data.

### 3.1 SELECTION MASKS

The selection masks allow to select parts of the data such as certain input channels or individual pixels of the different channels, see Figure 2. For each such mask, an associated cost can be defined, which can be used to adapt the masks to the specific requirements of the task at hand (e. g., if selecting pixels from one channel causes less data transfer in the inference phase than from another channel). Our optimization approach resorts to the following mask realizations, see Figure 3:

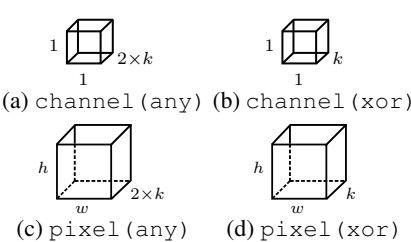

Figure 3: Implementation of masks

- *channel(any):* To select an arbitrary number of $k$ input channels, a joint mask $\boldsymbol{m}^D \in \{0,1\}^{1\times1\times k\times2}$ is used, which contains, for each of the $k$ channels, two weights. For instance, a mask $\boldsymbol{m}^D$ with $\boldsymbol{m}^D_{[1,1,1,:]} = (1,0)$ and $\boldsymbol{m}^D_{[1,1,2,:]} = (0,1)$ corresponds to selecting the first but not the second channel. Before applying the mask to an image $\boldsymbol{x} \in \mathbb{R}^{w\times h\times k}$, the first two axes are broadcasted, which yields a mask $\boldsymbol{m}^D \in \{0,1\}^{w\times h\times k\times2}$.
- *channel(xor):* In a similar fashion, one can select exactly one of the $k$ input channels by resorting to a joint mask of the form $\boldsymbol{m}^D \in \{0,1\}^{1\times1\times k}$. Here, exactly one of the $k$ weights equals one. For instance, a mask $\boldsymbol{m}^D$ with $\boldsymbol{m}^D_{[1,1,:]} = (0,0,\ldots,0,1)$ corresponds to only the last channel being selected. As before, the first two axes are broadcasted prior to the application of the mask, yielding a mask of the form $\boldsymbol{m}^D \in \{0,1\}^{w\times h\times k}$.
- *pixel(any):* To conduct pixel-wise selections, one can directly consider joint masks $\boldsymbol{m}^D \in \{0,1\}^{w\times h\times k\times2}$, which permit to select individual pixels per channel. For instance, a mask $\boldsymbol{m}^D$ with $\boldsymbol{m}^D_{[i,i,1,:]} = (1,0)$ and $\boldsymbol{m}^D_{[i,i,2,:]} = (1,0)$ for $i = 1,\ldots,w$ corresponds to selecting all pixels on the diagonal for the first two channels.
- *pixel(xor):* Similarly, one can only allow one channel to be selected per pixel by considering a joint mask of the form $\boldsymbol{m}^D \in \{0,1\}^{w\times h\times k}$, which contains, for each pixel, exactly one non-zero element corresponding to the selected channel for that pixel.

Note that variants of these four selection schemes can easily be obtained. For instance, shapes can be defined that partition the input data into, say, nine rectangular cells by considering masks of the form $\boldsymbol{m}^D \in \{0,1\}^{3\times3\times k\times2}$, where the first two axes are broadcasted to the corresponding cells. Such variants would allow to select certain cutouts, see Figure 4. The particular masks can be chosen according to the specific transfer capabilities between server and client. Finally, the different selection masks can also be applied sequentially with individual costs being assigned to them, see Section 4.

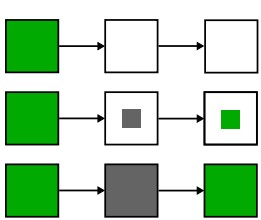

Figure 4: block(any)

---

**Algorithm 1:** LearnSelectionMasks($f, T$)

---

**Input :** model $f$ and training set $T$

1   $\boldsymbol{m} \leftarrow$ InitAllMasks()                  // initialize all selection masks
2   $\lambda, \tau \leftarrow$ InitLambdaTau()               // initialize lambda and tau
3   **for** $i \leftarrow 1$ **to** $n_{epoch}$ **do**
4      **for** $j \leftarrow 1$ **to** $n_{batch}$ **do**
5          $\boldsymbol{x}, y \leftarrow$ GetBatch($T$)                  // get next batch
6          $b \leftarrow j \mod 2 = 0$      // alternate between exploration/fixation
7          $\boldsymbol{m}^D, \boldsymbol{m}^S \leftarrow$ DiscretizeMasks($\boldsymbol{m}, \tau, b$)         // compute masks
8          $\hat{\boldsymbol{x}} \leftarrow$ ApplyMasks($\boldsymbol{x}, \boldsymbol{m}^D$)          // apply masks to input
9          $\hat{y} \leftarrow f(\hat{\boldsymbol{x}})$                 // compute prediction
10          $\mathcal{L} \leftarrow \mathcal{L}_f(\hat{y}, y) + \lambda \mathcal{Q}(\boldsymbol{m}^D)$         // compute adapted loss
11          $f, \boldsymbol{m} \leftarrow$ Optimize($f, \boldsymbol{m}, \boldsymbol{m}^S, \mathcal{L}$) // update weights of masks and model
12      $\lambda, \tau \leftarrow$ AdaptLambdaTau($\lambda, \tau$)          // adapt lambda and tau
13   $\boldsymbol{m}^D \leftarrow$ DiscretizeMasks($\boldsymbol{m}, \tau, false$)        // extract discretized masks
14   **return** $f, \boldsymbol{m}^D$

---

### 3.2   ALGORITHMIC FRAMEWORK

Let $T = \{(\boldsymbol{x}_1, y_1), \ldots, (\boldsymbol{x}_n, y_n)\} \subset X \times Y$ be a training set consisting of images $\boldsymbol{x}_i \in \mathbb{R}^{w \times h \times c}$ with associated labels $y_i \in \mathbb{R}$. The goal of the training process is to find suitable weight assignments both for the selection masks as well as for the neural network $f : X \rightarrow Y$ that is applied to the data.

#### 3.2.1   OPTIMIZATION APPROACH

Our procedure for learning suitable mask and network weights is given by LearnSelectionMasks, see Algorithm 1: Both the joint selection mask as well as the parameters $\lambda$ and $\tau$ are initialized in Line 1 and Line 2, respectively. The parameter $\lambda$ determines the trade-off between the task loss $\mathcal{L}_f$ and the mask loss $\mathcal{Q}$. Typically, $\lambda$ is initialized with a small positive value (e. g., $\lambda = 0.1$) and is gradually increased during training. Both the selection mask $\boldsymbol{m}$ and the network $f$ are trained simultaneously by iterating over a pre-defined number $n_{\text{epoch}}$ of epochs, each being split into $n_{\text{batch}}$ batches (for the sake of exposition, we assume a batch size of 1). For each batch, a discrete mask $\boldsymbol{m}^D$ is computed via the procedure DiscretizeMasks, which is used to obtain the masked image $\hat{\boldsymbol{x}}$. The induced prediction $\hat{y}$ is then used to compute the task loss $\mathcal{L}_f(\hat{y}, y)$. In addition, the overall mask loss $\mathcal{Q}(\boldsymbol{m}^D)$ is computed. Note that the discretized weights $\boldsymbol{m}^D$ are used in the forward pass, whereas a mask $\boldsymbol{m}^S$ with real-valued weights is used in the backward pass in Line 11. After each epoch, both $\lambda$ and $\tau$ are adapted. As detailed below, the procedure DiscretizeMasks alternates between an "exploration" and a "fixation" phase, specified by the parameter $b$. The final discrete weights for the joint mask are computed in Line 13.

**Learning Discrete Masks:**   Naturally, exhaustive search schemes that find the optimal discrete weights by testing out all possible assignments are computationally infeasible. Simple greedy approaches such as forward/backward selection of channels become computationally very demanding and are thus ill-suited for pixel-wise selections. Learning such discrete masks is difficult since the induced objective is not differentiable, which rules out the use of gradient-based optimizers commonly applied for training neural networks. One way to circumvent this problem is the so-called *Gumbel-Max trick*, which has been recently proposed in the context of variational auto-encoders to learn discrete latent variables (Maddison et al., 2014; Gumbel, 1954; Jang et al., 2017).[3] The procedure DiscretizeMasks uses this trick to discretize the masks $\boldsymbol{m}$, which contain class probabilities, in the forward pass of Algorithm 1. For instance, given a mask $\boldsymbol{m} \in \mathbb{R}^{1 \times 1 \times k \times 2}$ corresponding to channel(any), the procedure yields a discrete mask $\boldsymbol{m}^D \in \{0, 1\}^{1 \times 1 \times k \times 2}$ via

$$\boldsymbol{m}^D_{[1,1,j,:]} = \text{one\_hot} \left( \underset{i \in \{1,2\}}{\arg\max} \, \log \boldsymbol{m}_{[1,1,j,i]} + g_i \right) \tag{1}$$

---

[3]The Gumbel-Max trick yielded better results compared to other discretization methods such as L2-regularization along with a truncation of small weights.

where $j \in \{1, \ldots, k\}$ corresponds to the $j$-th channel and where each $g_i$ is either zero or a sample from the Gumbel distribution, depending on which phase is executed (see below). Equation (1) does not provide gradient information because $\arg\max$ cannot be differentiated. For this reason, the following differentiable surrogate $\boldsymbol{m}^S \in \mathbb{R}^{1 \times 1 \times k \times 2}$ is employed for the backward pass in Line 11:

$$\boldsymbol{m}_{[1,1,j,:]}^S = \texttt{softmax} \left( \frac{\log \boldsymbol{m}_{[1,1,j,1]} + g_1}{\tau}, \frac{\log \boldsymbol{m}_{[1,1,j,2]} + g_2}{\tau} \right) \tag{2}$$

Thus, the $\texttt{softmax}$ function is used as a surrogate for the discrete $\arg\max$ operation. The parameter $\tau$ is called *temperature*. A large $\tau$ leads to the resulting weights being close to uniformly distributed, whereas a small value for $\tau$ renders the values outputted by the $\texttt{softmax}$ surrogate being close to the discrete one-hot encoded vectors. The procedure $\texttt{DiscretizeMasks}$ alternates between "explore" and "fixate", specified by the parameter $b$. If $b$ is true, then $g_i$ is a random sample from the Gumbel distribution $g_i = -\log\left(-\log\left(u\right)\right)$ with uniform sample $u \sim U(0,1)$. If $b$ is false, $g_i = 0$. In the exploration phase, the optimizer can try out new possible mask assignments, whereas the network weights are adapted to the new data input in the fixation phase. The amount of changes made during the exploration phase is also influenced by the temperature parameter $\tau$.

**Initialization and Adaptation:**   The selection goal influences the initialization of the mask $\boldsymbol{m}$. In case all input channels for the $\texttt{channel(any)}$ scheme are equally important, the individual masks are set to $\boldsymbol{m}_{[1,1,j,:]} = (1 + \varepsilon, 0 + \varepsilon)$ for all $j = 1, \ldots, k$ to initially "select" all of the channels, where $\varepsilon \sim \mathcal{N}(0, \sigma)$ for some small $\sigma > 0$. In case the channels should be treated differently, the initialization can be adapted accordingly. For instance, only the first channel can be selected initially by setting $\boldsymbol{m}_{[1,1,j,:]} = (1 + \varepsilon, 0 + \varepsilon)$ for $j = 1$ and $\boldsymbol{m}_{[1,1,j,:]} = (0 + \varepsilon, 1 + \varepsilon)$ for $j \neq 1$.

The procedure $\texttt{InitLambdaTau}$ initializes both $\lambda$ and $\tau$. The parameter $\lambda$, which determines the trade-off between the task loss $\mathcal{L}_f$ and the loss $\mathcal{Q}$ associated with all masks, is initialized to a small value (e.g., $\lambda = 0.1$). The temperature parameter $\tau$ is initialized to a positive constant $\tau_{init}$ (e.g., $\tau_{init} = 10$). The adaptation of both $\lambda$ and $\tau$ after each epoch are handled by the procedure $\texttt{AdaptLambdaTau}$: In the course of the training process, the influence of $\lambda$ is gradually increased until $n_{\text{epoch}}$ epochs have been processed or some other stopping criterion is met (e.g., as soon as the desired reduction w.r.t. $\mathcal{Q}$ is achieved). Since the range of values for the model loss $\mathcal{L}_f$ is generally not known beforehand, we resort to a scheduler that increases $\lambda$ in Line 10 of Algorithm 1 in case the overall error $\mathcal{L} = \mathcal{L}_f + \lambda\mathcal{Q}$ has not decreased for a certain amount of epochs. The scheduler behaves similarly to standard learning schedulers, but instead of decreasing the learning rate, the value for $\lambda$ is increased by a certain factor $\lambda_{fac}$ (e.g., $\lambda_{fac} = 1.1$). The temperature $\tau$ influences the outcome of the $\texttt{softmax}$ operation in Equation (2): A large value leads to similar weights being mapped to similar ones via the operation, whereas a small value for $\tau$ amplifies small differences such that the outputted weights $\boldsymbol{m}^S$ are close to zero/one. For each new assignment of $\lambda$, we resort to some cool-down sequence, where $\tau$ is reset to $\tau = \tau_{init}$ and gradually decreased by a factor $\tau_{decay}$ after each epoch (e.g., $\tau_{decay} = 0.9$). This cool-down sequence let the process explore different weight assignments at the beginning, whereas binary decisions are fostered towards the end.

### 3.3   Extension and Reduction

Different costs can be assigned to the individual masks, which are jointly taken into account by the overall mask loss $\mathcal{Q}(\boldsymbol{m}^D)$. For instance, given $k$ input channels, one can resort to different losses $\mathcal{Q}_1, \ldots, \mathcal{Q}_k$ to favor the selection of certain channels. This turns out to be useful in case different "versions" for the input channels are

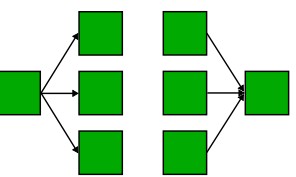

Figure 5: $\texttt{extend}$/$\texttt{merge}$

available, whose transfer costs vary (e.g., compressed images or thumbnails of different sizes). Often, pre-trained networks with a fixed input structure are given. The selection of different versions for such networks can be handled via simple operators, see Figure 5: The $\texttt{extend}$ operator can be used to extend a given input feature map (e.g., by generating ten compressed versions of different quality), whereas the $\texttt{merge}$ operator can combine feature maps in a user-defined way (e.g., by summing up the input channels). For instance, an $\texttt{extend}$ operation followed by a $\texttt{channel(xor)}$ selection and a $\texttt{merge}$ operation can be used to gradually select a certain version of each input channel *without* significantly changing the input for a given network in each step, thus allowing to learn masks for pre-trained networks without having to retrain the network weights from scratch, see Section 4.

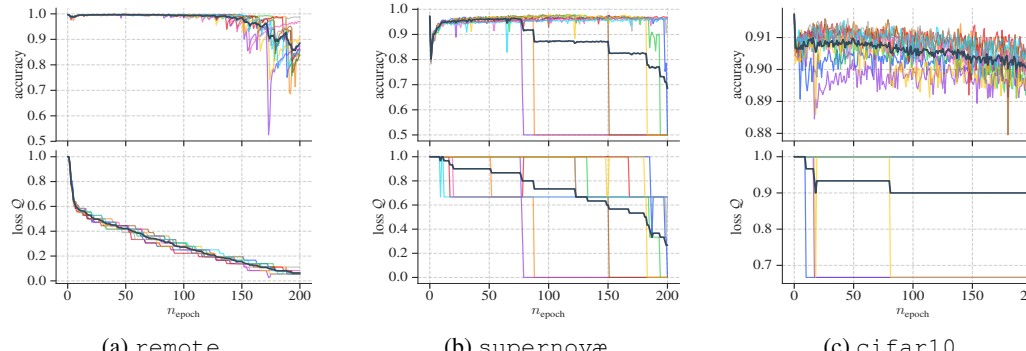

(a) remote      (b) supernovæ      (c) cifar10

Figure 6: channel(any) mask realization results on remote, supernovæ, and cifar10. The black line is the average value of the runs and individual runs are displayed in different colors.

## 4 EXPERIMENTS

Table 1: Datasets and Models

| Dataset | #train | #test | #class | $w$ | $h$ | $c$ | model |
|---|---|---|---|---|---|---|---|
| remote | 24 694 | 24 694 | 12 | 35 | 35 | 36 | AllConvNet |
| supernovæ | 4 020 | 4 018 | 2 | 50 | 50 | 3 | AllConvNet |
| cifar10 | 50 000 | 10 000 | 10 | 32 | 32 | 3 | ResNet101 |
| mnist | 60 000 | 10 000 | 10 | 28 | 28 | 1 | LeNet5 |
| svhn | 73 257 | 26 032 | 10 | 32 | 32 | 3 | ResNet101 |

We implemented our approach in Python 3.6 using PyTorch (version 1.1). Except for the trade-off parameter $\lambda$, default parameters were used for all experiments ($n_{batch} = 128$, $\tau_{init} = 10$, $\tau_{decay} = 0.5$, and $\tau_{min} = 0.01$). The learning rates $\beta$ for all selection masks were set to $\beta = 0.01$. For the networks, the Adam (Kingma & Ba, 2014) optimizer with AMSGrad (Reddi et al., 2018) and learning rate 0.0001 was used. The initial assignment $\lambda_{init}$ as well as the factor $\lambda_{fac}$ for $\lambda$ can have a significant impact. For this reason, we considered a small grid $(\lambda_{init}, \lambda_{fac}) \in \{0.1, 1.0\} \times \{1.1, 1.25\}$ of possible assignments. The influence of this parameter is shown in Figure 14; for all other figures, one of the four configurations is presented.

We considered several classification datasets and network architectures, see Table 1. In addition to the well-known cifar10, mnist, and svhn datasets (Krizhevsky et al., 2009; LeCun et al., 2010; Netzer et al., 2011), we considered two datasets from remote sensing and astronomy, respectively. For each instance of remote, one is given an image with 36 channels originating from six multi-spectral bands available for six different dates (Prishchepov et al., 2012). The learning goal is to predict the type of change occurring in the central pixel of each image. The astronomical dataset is related to detecting supernovæ (Scalzo et al., 2017). Each instance is represented by an image with three channels and the goal is to predict the type of object in the center of the image (a balanced version of the dataset was used). Both remote and supernovæ depict typical datasets in remote sensing and astronomy, respectively, with the target objects being located in the centers of the images. For all experiments, we considered a fixed amount of epochs and monitored the classification accuracy on the hold-out set. Each experiment was conducted $n_{runs} = 10$ times and the lines of the figures represent individual runs (the thicker black line is the aggregated mean over all runs). If not stated otherwise, we considered pre-trained networks before applying our selection approach.

### 4.1 CHANNEL SELECTION

The first experiment addressed the task of selecting a subset of the input channels. We used remote, supernovæ, and cifar10 as datasets, for which different outcomes were expected. For each of the $c$ channels, we assigned the same mask loss $\mathcal{Q}_i = 1/c$. The overall mask loss $\mathcal{Q}$ was the sum over all channels, which corresponds to the ratio of the data that need to be transferred. The outcome is shown in Figure 6. As expected, channel-wise selection worked best on remote due to many channels carrying similar information. Only if less than 20% of the channels were selected, the accuracy started to drop. In Figure 7, the selection process is sketched, where each row represents a different epoch (from top to bottom: 0, 50, 100, 150, 200) and where each columns corresponds to one of the channels. For supernovæ, the removal of a single channel did not significantly affect the classification accuracy. For some runs, all channels were removed at once, which indicates that the steps made for $\lambda$ were too large (thus, a smaller $\lambda_{fac}$ should be considered).

Figure 7: Selected channels for remote

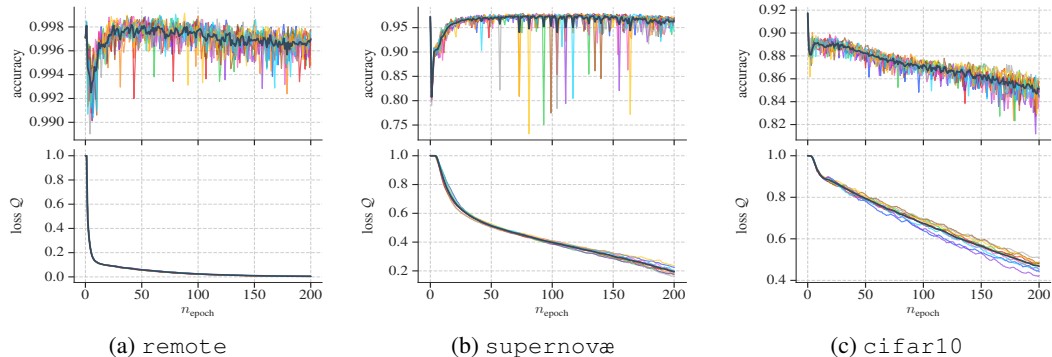

(a) `remote`        (b) `supernovæ`        (c) `cifar10`

Figure 9: `pixel(any)` mask realization results on `remote`, `supernovæ`, and `cifar10`.

On `cifar10`, only one of the three channels could be dropped with a minimal degradation of accuracy. Thus, as expected, less channels could be removed for both `supernovæ` and `cifar10` due to the channels being less redundant.

## 4.2 PIXEL-WISE SELECTION

Next, pixel-wise selections were addressed (`pixel(any)`) by conducting a similar experiment using the same datasets. The mask loss $\mathcal{Q}$ was obtained by summing over the selected pixels, where a weight of $1/w \times h \times c$ was assigned to each individual pixel. Thus, $\mathcal{Q}$ corresponds to the ratio of pixels that need to be transferred. The results are given in Figure 9. It can be seen that all plots for $\mathcal{Q}$ are smoother than for the channel-wise selections, which is due to the fact that the selection decisions to be made at each step were much more fine-grained (for `cifar10` and `supernovæ`, only three channels but thousands of subpixels are given). It can be seen that the accuracy drops slightly at the beginning of the training process. This is because the networks were not trained with missing inputs before and, hence, had to learn to compensate the missing input at the beginning. This effect could be lessened by (a) adding dropout layers to the networks or by (b) decreasing both $\lambda_{init}$ and $\lambda_{fac}$ to let the approach do less exploration at the beginning. Overall, the achieved reduction w.r.t. the remained accuracy is higher than for the channel-wise selection, although there are notable spikes in `supernovæ` that most likely stem from the removal of subpixels being crucial for the classification task (the removal of some central pixels seem to have had a significant impact). The development of the masks w.r.t. $n_{\text{epoch}}$ is shown in Figure 8 for `supernovæ`.

Figure 8: Pixel-wise selections

## 4.3 FEATURE MAP SELECTION

In many cases, preprocessed data are available on the server/client side. The next experiment was dedicated to such scenarios. In particular, we considered ten compressed versions for the `cifar10` images of different JPEG qualities $q \in \{100, 95, 85, \ldots, 25, 15\}$. The goal was to select one of these versions via `channel(xor)`. To capture the varying costs for the transfer of the different versions, we assigned $\mathcal{Q}_q = q/c \cdot 100$ to each version with quality level $q$. This mask loss is not as directly linked to the transfer costs, as the individual JPEG levels can have different impacts on each image, which depends on the JPEG compression algorithm. Also, the masks were initialized in such a way that only the version with the highest quality was selected initially.

Figure 10: JPEG on `cifar10`

Figure 10 shows the results. It can be seen that the lowest possible value (0.15) was obtained for $\mathcal{Q}$, for which an accuracy of about 82% remained. Also, an accuracy of about 88% could be maintained while reaching a loss of about $\mathcal{Q} \approx 0.5$. An illustration of the reduced input over the epochs is given in Figure 11.

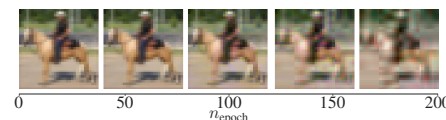

Figure 11: Reduced images

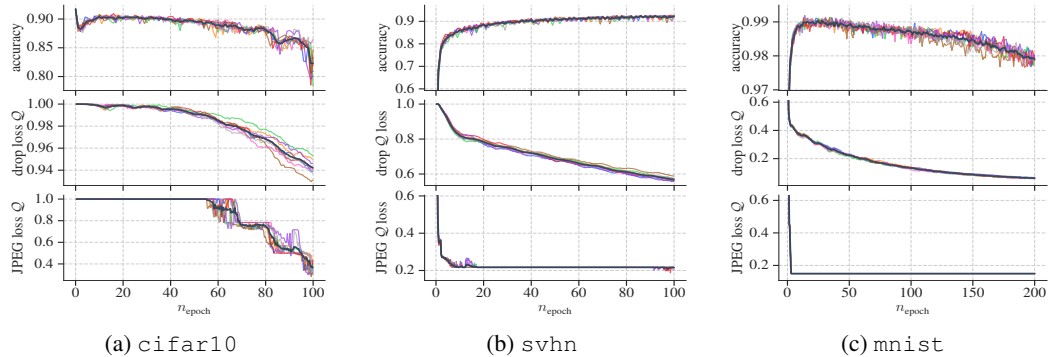

(a) `cifar10`          (b) `svhn`          (c) `mnist`

Figure 13: Results for the combination of selection masks on `cifar10`, `svhn`, and `mnist`, where JPEG qualities for each channel were used and, at the same time, pixels could be selected.

### 4.4 COMBINATION OF MASKS

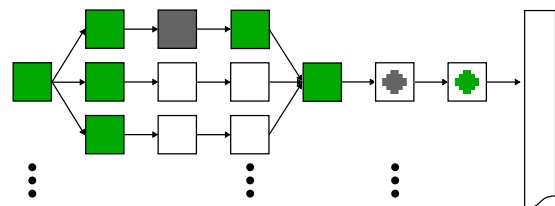

Next, multiple selection masks and mask losses were considered. The following operations were applied, see Figure 12: First, an `extend` operation was used to generate different JPEG qualities for each channel. Afterwards, a `channel(xor)` selection operation was applied, followed by a `merge` operation (sum). Finally, a `pixel(any)` selection was conducted to select subpixels of the merged channels. For this experiment, we used `cifar10`, `mnist`, and `svhn`. The joint mask loss $\mathcal{Q}$ was set to the product $q/c \cdot 100 \cdot 1/w \times h \times c$ of the two previously defined losses. The results are shown in Figure 13. Note that the models for `svhn` and `mnist` were not pre-trained in this case, which is why the accuracies start with a lower value. Since `mnist` is a dataset with many empty border pixels, our approach was able to remove 50% of the pixels in the first few epochs. Also, the lowest possible JPEG quality was used. Similar effects can be observed on `svhn` although it seems that is was harder to remove pixels due to more background pixels compared to `mnist`. For `cifar10`, the results show that the combined masks yielded similar outcomes as for the individual masks, see again Figure 9 and 10.

Figure 12: Combination of multiple selection masks.

### 4.5 INFLUENCE OF $\lambda$

The parameter $\lambda$ can have a big impact on the selection process. Figure 14 shows the influence of the four different configurations considered for our experiments given the `remote` dataset. It can be seen that a large $\lambda_{init}$ (blue and red line) leads to the mask loss $\mathcal{Q}$ quickly decreasing. For such settings, it seems that the network was not able to compensate the loss in information, which is why the accuracy was lower until the network was able to adapt to the new input. A smaller initial value for $\lambda$ leads to the selection process taking less input data away at the beginning, which avoids an initial drop of accuracy. Similarly, a large $\lambda_{fac}$ leads to a faster decrease w.r.t. $\mathcal{Q}$, which can be suboptimal in certain cases.

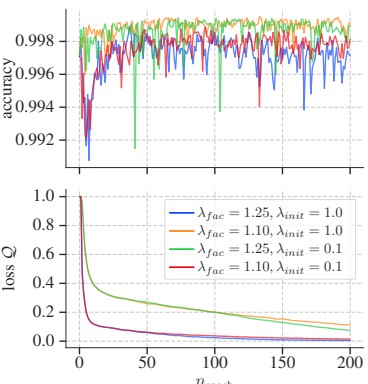

Figure 14: Influence of $\lambda$

### 5 CONCLUSIONS

The transfer of data between servers and clients can become a major bottleneck during the inference phase of a neural network. We propose a framework that allows to automatically select those parts of the data needed by the network to perform well, while, at the same time, minimizing the amount of selected data. Our approach resorts to various types of selection masks that are jointly optimized together with the corresponding network during the training phase. Our experiments show that it is often possible to achieve a good accuracy with significantly less input data needed to be transferred. We expect that such selection masks will play an important role for data-intensive domains such as remote sensing or astrophysics and for scenarios where the data transfer bandwidth is very limited.

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

## A    APPENDIX

### A.1    RATIO OF FIXATION/EXPLORATION

We introduced the fixation phase to allow the network to adapt to the mask changes made during the exploration phase. This can be seen as intermediate "post-training" to ensure that the "optimal" result is obtained for a given $\lambda$ (which is increased over time). The alternation between the exploration and fixation phase worked well in practice. However, other ratios between these two phases are also possible. This can be achieved by resorting to a corresponding ratio $\gamma$ in Line 6 of Algorithm 1. Here, $\gamma = 0.25$ corresponds to 1 fixation and 3 exploration iteration(s) and $\gamma = 0.75$ to 3 fixation and 1 exploration iteration(s).

The influence of the ratio $\gamma$ between exploration and fixation is shown in Figure 15 (which depicts an extension of Figure 9c). The ratio $\gamma$ does have the expected effect on the results. In particular, a larger value (more intermediate "post-training") yields slightly better accuracies (top) and slightly larger values for the reduction

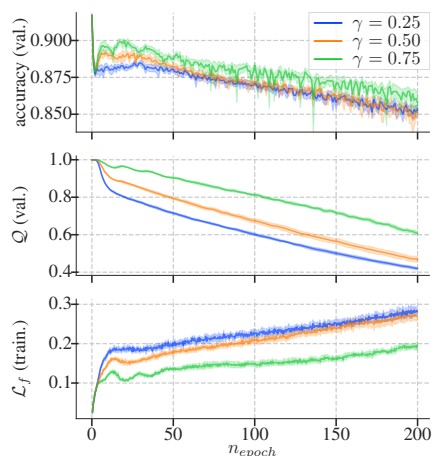

Figure 15: Influence of the ratio between exploration and fixation during training.

loss $\mathcal{Q}$ (middle). Hence, the ratio $\gamma$ can be used to influence speed up/slow down the pruning. We decided to omit this from the original algorithm to keep it simple since a similar effect can be achieved by adjusting the trade-off parameter $\lambda$ and the temperature $\tau$.

### A.2    DECREASE IN ACCURACY

One might question if the decrease in accuracy observed in the experiments result from the generalization gap, i. e., that the networks are simply overfitting. However, a steady decrease in accuracy is expected since more and more input data are masked out. Eventually, an accuracy close to the mean estimator will be obtained, since the mask and network weights are adapted according to the joint objective $\mathcal{L}_f + \lambda\mathcal{Q}$ and since $\lambda$ is increased in the course of the training process. Figure 15 (bottom) shows the training loss $\mathcal{L}_f$ (cross entropy) for the experiment described above. It can be seen that the training loss $\mathcal{L}_f$ increases as the validation accuracy decreases, which is a strong indicator that overfitting is not responsible for the decrease in accuracy (but the loss of information due to the mask changes). Note that the slope of the initial drop/increase (first 20 epochs) depends on the assignment for $\lambda_{init}$; here, smaller values for $\lambda_{init}$ lead to less changes at the beginning (see again Figure 14). Finally, it is worth mentioning that our approach is very stable w.r.t. the involved parameters (note that all hyper-parameters except for $\lambda$ were fixed).

### A.3    STOPPING CRITERIA

Instead of using a fixed number $n_{epoch}$ of epochs, other stopping criteria can also be used. For example, one could stop training the mask and the model as soon as the loss $\mathcal{Q}$ for the masks falls below a particular user-defined threshold or as soon as the accuracy has decreased significantly. Once the general training procedure has stopped, one could keep on adapting both the mask weights and the network weights for several epochs without increasing $\lambda$ any further, i. e., without changing $\mathcal{L}$ anymore. In addition, one could "finalize" the model $f$ by training the model (but not the mask) for several epochs.

