# OpenReview forum: "End-To-End Input Selection for Deep Neural Networks"
_ICLR.cc/2020/Conference — Reject_

### Official Review · AnonReviewer1 · 2019-10-23
**Official Blind Review #1**

**Rating:** 3

**Review:**

The authors argue that data transfer costs for ML inference may be significant in some scenarios, such as for remote sensing. They propose to jointly train a model to maintain good performance while significantly reducing input size by applying a global mask. To allow training with discrete masks, the Gumbel-softmax trick is used. The experiments cover multiple datasets and multiple types of mask. Generally, performance remains reasonably high and degrades gracefully as the input is increasingly masked.

I lean slightly towards rejecting the paper. The problem is interesting and potentially important, but many experiments are  too simplistic and lack strong baselines.

I believe the problem is well motivated, but not very much explored yet. There has been much work on reduce ML system computation costs and memory storage requirements, but mostly by modifying the model instead of the data. The paper proposes a reasonable approach to reduce data transfer costs.

The authors propose 4 types of masks (channel/any, channel/xor, pixel/any, pixel/xor), which are applicable under different circumstances. Some of them may also be combined. To learn discrete masks, they apply the Gumbel-softmax trick. Results clearly show that learning discrete masks (reducing input size) while maintaining decent performance is feasible .

As the objective function is modified during training by adjusting \lambda, the performance/size trade-off is only loosely specified. All presented results are learning curves, but there are no clear final numbers.

The channel selection task (4.1) is potentially interesting, but lacks a baseline. How does random selection of channels perform?

The pixel selection task (4.2) is simplistic. Using a cloud of pixels near the center of the images appears sufficient, which could be inferred by looking at a few samples and doesn't necessarily necessitate learning. Could the approach be extended to more complex images, predicting one mask per image instead of a global mask?

Feature map selection (4.3, channel 'xor') could be likely solved with hyper-parameter search, especially if the number of channels is small. Section 4.4 combines the previous two subsections, and it is unclear how much we gain from learning the masks over using simple heuristics.

More minor points:

In the related work section, the author could additionally mention distillation.

The Gumbel noise is used on half of the inputs, while the current argmax is used otherwise. It is unclear whether this is necessary, and there are no related experiments.

Although this is not crucial, for the 'any' variants, the last mask dimension appears superfluous (2-class softmax). The binary variant of Gumbel-softmax (Maddison et al. The Concrete Distribution: A Continuous Relaxation of Discrete Random Variables, Appendix B) could be used.

**Experience Assessment:**

I have read many papers in this area.

**Review Assessment: Checking Correctness Of Derivations And Theory:**

I assessed the sensibility of the derivations and theory.

**Review Assessment: Checking Correctness Of Experiments:**

I assessed the sensibility of the experiments.

**Review Assessment: Thoroughness In Paper Reading:**

I read the paper at least twice and used my best judgement in assessing the paper.

---

> ### Author Response · Authors · 2019-11-13
> **Rebuttal for Review #1**
>
> Thank you very much for your comments and suggestions and apologies for the slight delay; our response has been delayed due to personal reasons.
>
>
> Concern:
> “All presented results are learning curves, but there are no clear final numbers.”
>
> Answer:
> This is on purpose, since we do not want to choose a specific amount of features/options to select, but rather optimize and then pick the trade-off that the practitioner deems reasonable (and maybe fixate the model for that state a bit more).
>
>
> Concern:
> “How does random selection of channels perform?”
>
> Answer:
> We did random selection as a preliminary study and it performed worse with decreasing number of selected channels (for channel-wise selection) and a high number of channels. We have included these results in the updated version of our manuscript.
>
>
> Concern:
> “The pixel selection task (4.2) is simplistic.“
>
> Answer:
> The pixel selection mask allows to select the optimal amount (!) of pixels needed for a given task; typically, it is not clear a priori how large such a cloud of pixels has to be, and our framework allows to optimize for this size.
> Further, In this paper we focus on a static mask after training and no individual mask for each instance (since no further calculation are possible on the data server). Nevertheless, future work could include this option.
>
>
> Concern:
> “Feature map selection (4.3, channel 'xor') could be likely solved with hyper-parameter search”
>
> Answer:
> The channel selection task is most important for non-standard imagery, for example, satellite images with many channels or selection of different quality levels (select different compression algorithms). Another advantage is that we do not need to perform hyperparameter searches that might not account for dependencies between choices (e.g., if we take away one channel, we might take a complete other set of pixel, or vise versa).
>
>
> Concern:
> “the author could additionally mention distillation”
>
> Answer:
> We added distilling of networks to our references, although we are more concerned with the input masking and not with the training of the predictor network (this could be used together though).
>
>
> Concern:
> “Gumbel noise is only applied to half the inputs”
>
> Answer:
> Could you please specify this? We apply Gumbel noise to all selections while training in the exploration phase.
>
>
> Concern:
> “The binary variant of Gumbel-softmax (…) could be used”
>
> Answer:
> The binary variant is used in implementation for the ‘any’ masks, but we wanted to keep the manuscript coherent since it is theoretically the same.

---

### Official Review · AnonReviewer3 · 2019-10-23
**Official Blind Review #3**

**Rating:** 3

**Review:**

The paper addresses the problem of high-cost transfer between server and user for machine learning applications. The method proposes to augment inputs with channel/spatial masks that are trained via the Gumbel-softmax trick together with the model's weights trained/finetuned to account for the loss of information. These learned masks are then applied to the image before sending it to a server with where inference takes place to reduce file transfer costs. In the experimental study, the paper shows that on computer vision tasks inputs can be reduced with relatively little drop in accuracy and analyses how hyperparameters of the model affect its performance. The framework is also adapted to the downstream task-guided choice of compression techniques, e.g. which compression quality to choose for JPEG that will be an optimal trade-off in terms of downstream task quality vs data transfer cost.
Overall, this paper could be a valid algorithmic contribution, however, I have concerns about how this method fares with others, resolving the following issues in the author response will likely increase the score.
- A discussion on the connection/comparison with representation learning and dimensionality reduction (VAEs, quantization, etc) would help improve the exposition of the paper and help define how and when the suggested method is more appropriate to use, as well as citations to the other lines of work in reducing the model size (knowledge distillation [1], tensor decomposition approaches [2], adaptive computation time techniques [3], etc).
- I would like to see an experiment that compares other works mentioned as related work. [4] has been mentioned as one of the nontrivial compression methods for image data, how does the proposed method compare to it?
- Another experiment comparing a method from representation learning, e.g. a VAE trained with the embedding size corresponding to some optimal Q value in this work would be helpful.
- Please include reference accuracy values for the dataset/NN pairs used in Table 1.
- In Section 4.5, the paper states that large lambda values correspond to blue and red lines, however in the corresponding figure large lambda values correspond to blue and orange lines which exhibit different behaviors, could you please clarify this?

References
[1] Hinton, Geoffrey, Oriol Vinyals, and Jeff Dean. "Distilling the knowledge in a neural network."
[2] Novikov, Alexander, et al. "Tensorizing neural networks." Advances in neural information processing systems. 2015.
[3] Figurnov, Michael, et al. "Spatially adaptive computation time for residual networks." Proceedings of the IEEE Conference on Computer Vision and Pattern Recognition. 2017.
[4] Jiang, Feng, et al. "An end-to-end compression framework based on convolutional neural networks." IEEE Transactions on Circuits and Systems for Video Technology 28.10 (2017): 3007-3018.

**Experience Assessment:**

I have read many papers in this area.

**Review Assessment: Checking Correctness Of Derivations And Theory:**

I carefully checked the derivations and theory.

**Review Assessment: Checking Correctness Of Experiments:**

I carefully checked the experiments.

**Review Assessment: Thoroughness In Paper Reading:**

I read the paper thoroughly.

---

> ### Author Response · Authors · 2019-11-13
> **Rebuttal for Review #3**
>
> Apologies for the slight delay; our response has been delayed due to personal reasons. Thank you very much for your insightful review.
>
>
> Concern:
> “A discussion on the connection/comparison with representation learning and dimensionality reduction (VAEs, quantization, etc) would help improve the exposition of the paper”
>
> Answer:
> We do representation learning as in the general term that we learn a selection of features or quality levels, but we assume that no additional computations, such as VAE or similar can be performed on the data server.
> We added a distinction between dim. red. and feature selection to the manuscript.
> In short:
> Dim. reduction (VAEs, PCA, etc.) yields features in a lower dimensionality, but such methods do not “save” on features since every feature is used even if only to a small amount and change the structure of the data (e.g. from image to vector data).
> In addition, these methods are unsupervised and not task depending compared to our framework.
> Feature selection takes a pre-defined subset of the features (e.g. k out of d), but our approach decreases this k on its own and we can pick the best trade-off between prediction performance and feature cost.
> It is worth stressing that our framework jointly learns a selection of the data without changing its structure and the necessary adaptation to given neural network structure (original networks can be reused).
>
>
> Concern:
>  “I would like to see an experiment that compares other works mentioned as related work. [4] has been mentioned as one of the nontrivial compression methods for image data, how does the proposed method compare to it?”
>
> Answer:
> The work in [4] is a compression framework, whereas our approach is feature selection, which makes a comparison infeasible. That said, the representation from [4] could be another option to choose from when selecting pixel (e.g. choices could be RGB from original, JPEG, and [4]), one only needs to assign weights to the representations and check if the the compute resources are available on the data server (which we do not assume).
>
>
> Concern:
> “Please include reference accuracy values for the dataset/NN pairs used in Table 1.”
>
> Answer:
> We haven’t added those numbers yet due to space constraints, but this information is present in all plots as the initial evaluation (epoch 0) is done with the original network without any masking (loss Q=1).
>
>
> Concern:
> “In Section 4.5, the paper states that large lambda values correspond to blue and red lines …”
>
> Answer:
> Thank you for point out this issue! Yes, there was a mix-up in the plot, which we have have already adjusted.

---

### Official Review · AnonReviewer2 · 2019-10-24
**Official Blind Review #2**

**Rating:** 3

**Review:**

The paper presents an approach to discrete input selection for NNs, using the Gumbel-Softmax trick at its core. It motivates this problem in the context of communicating data over a network with limited bandwidth budget. It proposes constructing different kinds of masks that can be applied over channels or pixels in the input, grounding the discussion in the image domain. This can be seen as a special case of Feature Selection, with image specific substructures motivating the choice of mask types.

There is very little novelty in this work over that presented in Abubakar Abid et. al. [1], where the idea of using Gumbel-Softmax as a differentiable Feature Selection algorithm has already been expounded at depth, both in unsupervised as well as supervised settings. The current work draws directly from the supervised form in [1]. The only incremental contribution in this work is the specific mask types and mask-specific losses.

Pros
•	Interesting approach to extend the framework in [1] to CNNs, with use of masks and mask-specific loss
•	Clear motivation for the network bandwidth limited use case

Cons
•	Hardly any technical novelty because the core ideas are already presented as well as applied to the same task in [1]
•	It is very surprising that the authors do not even cite [1] in their paper, despite their work being extremely closely related to it
•	Most of the discussion in the Related Work section is unrelated to the specific task they tackled in the paper (i.e., input/feature selection). The second paragraph in this section talks about ‘gradient-driven search’ for discrete selection, which has been recently explored not only in [1] but also related G-S applications like [2], [3], but the authors seem unaware of this line of works
•	The authors do not compare their approach against any existing baselines from literature for this task, again with the most apt being [1] and baselines therein. This makes it hard to understand the true value of their proposals such as mask types, schedule that adjusts both ‘tau’ and ‘lambda’ during training etc.

[1] Abubakar Abid et al., “Concrete Autoencoders for Differentiable Feature Selection and Reconstruction”, ICML 2019, (https://arxiv.org/abs/1901.09346)
[2] Hanxiao Liu et al., “DARTS: Differentiable Architecture Search”, ICLR 2019
[3] Bichen Wu et al., “FBNet: Hardware-Aware Efficient ConvNet Design via Differentiable Neural Architecture Search” CVPR 2019


**Experience Assessment:**

I have read many papers in this area.

**Review Assessment: Checking Correctness Of Derivations And Theory:**

I assessed the sensibility of the derivations and theory.

**Review Assessment: Checking Correctness Of Experiments:**

I assessed the sensibility of the experiments.

**Review Assessment: Thoroughness In Paper Reading:**

I read the paper at least twice and used my best judgement in assessing the paper.

---

> ### Author Response · Authors · 2019-11-13
> **Rebuttal for Review #2**
>
> Apologies for the slight delay; our response has been delayed due to personal reasons. Thank you very much for your in-depth review and comments.
>
>
> Concern:
> “Hardly any technical novelty because the core ideas are already presented as well as applied to the same task in [1]. It is very surprising that the authors do not even cite [1] in their paper, despite their work being extremely closely related to it. Most of the discussion in the Related Work section is unrelated to the specific task”
>
> Answer:
>  We would like to stress that we were not aware of [1] at the time of finalizing our work, which we submitted to NeurIPS on May 23 this year (and which was considered borderline with scores of 6, 7, and 4). The corresponding draft was made available after the submission on Arxiv (using a slightly different title, which we do not reveal here due to the double blind review process).
>
> We have added [1] as well as [2] and [3] to the related work section of our manuscript. While [1] is related to our work, we would like to point out the following differences:
>
> Our framework resorts to selection masks, which can be adapted to the specific needs of the server/client capabilities. The masks can be used to enforce various additional constraints such as channel-wise selections. This is, in our opinion, a novel and original contribution.
> [1] requires the hyperparameter k to be pre-defined, while our approach is dynamic in a way that it reduces the amount of used features on its own, which could be more interesting if one is looking for the best performance/selection compromise.
> Since k is static, the network that follows cannot be (to our knowledge) convolutional, which limits the approach to fully connected networks. Furthermore, pre-trained networks are also not usable.
>
>
> Concern:
> “The authors do not compare their approach against any existing baselines from literature for this task, again with the most apt being [1] and baselines therein.”
>
> Answer:
> The two approaches are hard to compare since we do not select k features, but our mask model removes features on its own (maybe skipping k) going from all features or highest quality features (e.g. original image instead of jpeg version) to fewer and fewer.
> Furthermore, since we replace values not selected by zero, we are able to use pre-trained weights and convolutional networks instead of relying on fully connected networks that need to be trained from scratch and with no spatial information intact meaning the architectures are not necessarily comparable.  Nevertheless, we have conducted experiments on MNIST (not pre-trained weights) and compared the accuracy at k=50 (as in used in [1]) without any fixation phase and achieved 97.85% compared to the reported 90.6%. We would be happy to add these results as well as corresponding experiments on Fashion-MNIST.

---

> > ### Author Response · Authors · 2019-11-14
> > **New results for Fashion-MNIST**
> >
> > As mentioned, we also did the pixel-wise selection (any-mask) and stopped at 50 pixel (loss Q of 0.0638). We now have the results for Fashion-MNIST and without any fixation afterwards, we achieved 85.7% accuracy compared to 67.7% in [1].
> >
> > We would like to stress that this task only covers a subset of our frameworks capabilities (any-selection pixel-wise) and  a comparison to [1] is complicated:
> > - We are using an end-to-end supervised/task-dependent approach with an any-mask with 28x28 learnable parameters and an out-of-the-box LeNet5 (431080 learnable parameters) as classifier.
> > - [1] select  50 pixel (with no spatial information intact) in an unsupervised/task-independent way with an Auto-Encoder-like architecture and then trains a Random Forest as classifier on the selected features.

---

### Decision · Program_Chairs · 2019-12-19

**Decision:**

Reject

**Comment:**

This paper proposes to address the high bandwidth cost when transferring data between server and user for machine learning applications. The input data is augment with channel and spatial mask so that the file transfer cost is reduced. While the reviewers agree that this is a well motivated and interesting problem to study, a number of concerns are raised, including loosely specified performance/size trade-off, how this work is compared to related work, low novelty relative to a few key missing references. The authors respond to Reviewers’ concerns but did not change the rating. The ACs concur the concerns and the paper can not be accepted at its current state.